# Diagnostic Value of VEGF-A, VEGFR-1 and VEGFR-2 in Feline Mammary Carcinoma

**DOI:** 10.3390/cancers13010117

**Published:** 2021-01-01

**Authors:** Catarina Nascimento, Andreia Gameiro, João Ferreira, Jorge Correia, Fernando Ferreira

**Affiliations:** 1CIISA—Centro de Investigação Interdisciplinar em Sanidade Animal, Faculdade de Medicina Veterinária, Universidade de Lisboa, Avenida da Universidade Técnica, 1300-477 Lisboa, Portugal; catnasc@fmv.ulisboa.pt (C.N.); agameiro@fmv.ulisboa.pt (A.G.); jcorreia@fmv.ulisboa.pt (J.C.); 2Instituto de Medicina Molecular, Faculdade de Medicina, Universidade de Lisboa, 1649-028 Lisboa, Portugal; hjoao@medicina.ulisboa.pt

**Keywords:** feline mammary carcinoma, VEGF-A, VEGFR-1, VEGFR-2, non-invasive biomarkers, angiogenesis

## Abstract

**Simple Summary:**

Feline mammary carcinoma (FMC) is the third most common neoplasia in the cat, showing a highly malignant behavior, with both HER2-positive and triple negative (TN) subtypes presenting worse prognosis than luminal A and B subtypes. Furthermore, FMC has become a reliable cancer model for the study of human breast cancer, due to the similarities of clinicopathological, histopathological, and epidemiological features among the two species. Therefore, the identification of novel diagnostic biomarkers and therapeutic targets is needed to improve the clinical outcome of these patients. The aim of this study was to assess the potential of the VEGF-A/VEGFRs pathway, in order to validate future diagnostic and checkpoint-blocking therapies. Results showed that serum VEGF-A, VEGFR-1, and VEGFR-2 levels were significantly higher in cats with HER2-positive and TN normal-like tumors, presenting a positive association with its tumor-infiltrating lymphocytes expression, suggesting that these molecules may serve as promising non-invasive diagnostic biomarkers for these subtypes.

**Abstract:**

Vascular endothelial growth factor (VEGF-A) plays an essential role in tumor-associated angiogenesis, exerting its biological activity by binding and activating membrane receptors, as vascular endothelial growth factor receptor 1 and 2 (VEGFR-1, VEGFR-2). In this study, serum VEGF-A, VEGFR-1, and VEGFR-2 levels were quantified in 50 cats with mammary carcinoma and 14 healthy controls. The expression of these molecules in tumor-infiltrating lymphocytes (TILs) and in cancer cells was evaluated and compared with its serum levels. Results obtained showed that serum VEGF-A levels were significantly higher in cats with HER2-positive and Triple Negative (TN) Normal-Like subtypes, when compared to control group (*p* = 0.001, *p* = 0.020). Additionally, serum VEGFR-1 levels were significantly elevated in cats presenting luminal A, HER2-positive and TN Normal-Like tumors (*p* = 0.011, *p* = 0.048, *p* = 0.006), as serum VEGFR-2 levels (*p* = 0.010, *p* = 0.046, *p* = 0.005). Moreover, a positive interaction was found between the expression of VEGF-A, VEGFR-1, and VEGFR-2 in TILs and their serum levels (*p* = 0.002, *p* = 0.003, *p* = 0.003). In summary, these findings point to the usefulness of VEGF-A and its serum receptors assessment in clinical evaluation of cats with HER2-positive and TN Normal-Like tumors, suggesting that targeted therapies against these molecules may be effective for the treatment of these animals, as described in human breast cancer.

## 1. Introduction

Human breast cancer is the most diagnosed cancer and the leading cause of cancer-related death in women worldwide [1], being a heterogeneous disease driven by five distinct molecular profiles (Luminal A, Luminal B, HER2-positive, Triple-Negative Normal and Basal-Like) [2,3]. In parallel, the feline mammary carcinoma (FMC) is a very common neoplasia associated with local recurrence and distant metastasis, resulting in a high mortality rate [4], being HER2-positive and Triple Negative (TN) the most aggressive subtypes [5,6]. Furthermore, FMC has become a reliable cancer model for the study of human breast cancer, due to the similarities of clinicopathological, histopathological and epidemiological features among the two species [7,8,9]. Therefore, the development of new approaches allowing the early detection and appropriate therapeutic strategies and follow-up of cats with mammary carcinoma becomes crucial.

Angiogenesis, the formation of new blood vessels, is a hallmark of cancer and is fundamental to supply the high metabolic demands in nutrients and oxygen of cancer cells, leading to a rapid tumor growth and metastatic dissemination [10,11]. Accordingly, cancer cells and stromal cells are able to produce and release mediators of angiogenesis, such as the vascular endothelial growth factor A (VEGF-A) [12,13,14]. VEGF-A is a glycoprotein (45 kDa) that is highly conserved among mammalian species, being expressed by different cell populations, as tumor infiltrating lymphocytes (TILs), macrophages, platelets and cancer cells, promoting capillary network growth and vascular permeability, allowing cancer cells to migrate to distinct organs [15,16,17]. In humans, there are four distinct VEGF-A isoforms with 121, 165, 189, and 206 amino acids, as a result of alternative mRNA splicing, with VEGF_165_ being the predominant isoform [13,18,19]. Several studies in human breast cancer have shown that VEGF-A overexpression is present in tumors with aggressive phenotype, such as HER2-positive and TN subtypes [20,21], being associated with poor prognosis and shorter disease-free survival (DFS) and overall survival (OS) [11,16]. Nevertheless, to show its biological activity, this angiogenic cytokine needs to bind to specific class-III-membrane tyrosine kinase receptors expressed on endothelial cells, as the vascular endothelial growth factor receptor 1 and 2 (VEGFR-1/Flt-1; VEGFR-2/KDR/Flk-1) [22], both having seven extracellular immunoglobulin homology domains, a transmembrane domain and an intracellular region with a tyrosine kinase domain, leading to distinct biological effects [23]. Accordingly, VEGFR-1 is more related with the pathological angiogenesis, while VEGFR-2 is involved in physiological and pathological angiogenesis [13]. In humans, the interaction between VEGF-A and VEGFR-2 is the most relevant for angiogenesis in solid tumors [12], as VEGFR-2 binds to all VEGF-A isoforms [24]. Activated VEGFR-2 promotes the activation of the PLC-γ, PKC-Raf-1-MEK-MAP kinase and PI3K-AKT pathways, as a signaling towards cell proliferation and endothelial cell survival [24,25].

Furthermore, it has been described that the secretion of soluble forms of VEGFR-1 (sVEGFR-1) and VEGFR-2 (sVEGFR-2) in the extracellular matrix displayed high affinity to VEGF-A. These isoforms are considered a natural defense strategy against malignant cells, exhibiting antiangiogenic, anti-edema and anti-inflammatory effects [13,22]. Accordingly, a low sVEGFR-1/VEGF-A ratio was associated with higher tumor malignancy and poor prognosis [13].

The discovery of antitumor immunotherapies targeting tumor-induced angiogenesis (e.g., VEGF-A, VEGFR-2) have been proposed as a universal therapeutic strategy to improve the clinical outcome of patients with several solid tumor types, as breast cancer [18,26]. Studies demonstrated that a humanized monoclonal antibody that bind to all soluble VEGF-A isoforms, bevacizumab, inhibit angiogenesis and tumor growth, promoting significant improvements in DFS of patients with breast cancer [10,27]. However, an increased overall survival (OS) could not be demonstrated, leading to a bevacizumab’s approval withdrew by Food and Drug Administration (FDA) after two years following its initial approval, whereas the European Medicines Agency (EMA) maintained their approval [10,11]. Furthermore, several novel and potent VEGFR-1 and VEGFR-2 antagonists are being evaluated in clinical trials, showing promising results [24,28]. In cat, although Michishita et al. (2016) demonstrated that bevacizumab suppressed tumor growth in a xenograft model, suggesting its potential therapeutic effect for FMC [29], the role of VEGF-A in angiogenesis and its biological effects in feline mammary carcinoma is still poorly documented. Therefore, the aim of this study was to: (i) quantify and compare the serum VEGF-A, VEGFR-1 and VEGFR-2 levels between cats with distinct mammary carcinoma subtypes and healthy controls; (ii) test for associations between serum levels and clinicopathological features; (iii) evaluate the VEGF-A, VEGFR-1 and VEGFR-2 expression in TILs and cancer cells of feline spontaneous mammary carcinomas and (iv) screen for correlations between serum levels and expression levels of VEGF-A, VEGFR-1 and VEGFR-2 in TILs and cancer cells.

## 2. Results

### 2.1. Serum VEGF-A, VEGFR-1 and VEGFR-2 Levels Are Significantly Elevated in Cats with HER2-Positive and TN Normal-Like Mammary Carcinoma

Cats with mammary carcinoma were stratified according to their tumor subtype and serum VEGF-A, VEGFR-1 and VEGFR-2 levels were measured and compared with control group. Results showed that cats with HER2-positive and TN Normal-Like mammary carcinoma displayed higher serum VEGF-A levels than control group (1748.6 ± 3558.4 pg/mL vs. 0.0 pg/mL, *p* = 0.001; 1881.9 ± 2927.9 pg/mL vs. 0.0 pg/mL, *p* = 0.020; respectively, Figure 1A). Furthermore, cats presenting Luminal A, HER2-positive and TN Normal-Like mammary carcinoma subtypes revealed higher serum VEGFR-1 levels, comparing with healthy group (10197.4 ± 17679.4 pg/mL vs. 0.0 pg/mL, *p* = 0.011; 3068.9 ± 4935.5 pg/mL vs. 0.0 pg/mL, *p* = 0.048; 11527.6 ± 12845.4 vs. 0.0 pg/mL, *p* = 0.006; respectively, Figure 1B), as well as serum VEGFR-2 levels (2033.4 pg/mL ± 3550.7 vs. 0.0 pg/mL, *p* = 0.010; 502.3 ± 1091.8 pg/mL vs. 0.0 pg/mL, *p* = 0.046; 2023.6 ± 2416.0 pg/mL vs. 0.0 pg/mL, *p* = 0.005; respectively, Figure 1C).

In addition, a positive correlation was identified between serum VEGF-A and both VEGFR-1 (r = 0.567, *p* = 0.0001) and VEGFR-2 levels (r = 0.591, *p* = 0.0001), and also between serum VEGFR-1 and VEGFR-2 levels (r = 0.973, *p* = 0.0001).

### 2.2. Higher Serum VEGFR-1 and VEGFR-2 Levels are Correlated with the Administration of Contraceptives and Low-Grade Feline Mammary Carcinomas

A statistical analysis was performed between the serum VEGF-A, VEGFR-1 and VEGFR-2 levels in cats with mammary carcinoma and the studied clinicopathological features (Table 1). Although, no significant associations were found between serum VEGF-A levels and the recorded clinicopathologic parameters, serum VEGFR-1 and VEGFR-2 levels were positively associated with contraceptive administration (*p* = 0.026 and *p* = 0.042, respectively, Figure 2A,B) and tumors of lower malignancy grade (*p* = 0.037 and *p* = 0.046, respectively, Figure 2C,D).

### 2.3. Serum VEGF-A, VEGFR-1 and VEGFR-2 Levels Are Positively Associated with Their Expression in Tumor Infiltrating Lymphocytes

Regarding the above results, the expression of VEGF-A, VEGFR-1 and VEGFR-2 was analyzed in cancer cells and in tumor infiltrating lymphocytes (TILs). Accordingly, the immunostaining analysis of cancer cells revealed that 95% (70% weak positive; 25% strong positive), 19% (17% weak positive; 2% strong positive) and 19% (19% weak positive; 0% strong positive) of tumors showed a positive score for VEGF-A, VEGFR-1 and VEGFR-2, respectively. In addition, 51% (33% weak positive; 18% strong positive), 22% (22% weak positive; 0% strong positive) and 24% (21% weak positive; 3% strong positive) of the tumors showed a positive IHC staining in TILs for VEGF-A, VEGFR-1 and VEGFR-2. Moreover, VEGF-A (Figure 3A,B) and VEGFR-1 expression (Figure 3C,D) was detected in cytoplasm of both cell types, while VEGFR-2 expression (Figure 3E,F) was found in cytoplasm and nucleus.

Results also revealed that serum VEGF-A levels were significantly higher in cats showing a strong positive VEGF-A expression in TILs, in comparison to those with a weak positive (*p* = 0.003) or negative (*p* = 0.003) score (Figure 4A). Furthermore, a positive association was found between weak positive VEGFR-1 and VEGFR-2 expressions in TILs and their correspondent serum levels (*p* = 0.002, Figure 4B; *p* = 0.002, Figure 4C). No significant correlations were found between serum VEGF-A, VEGFR-1, or VEGFR-2 levels and the expression of these proteins in cancer cells (*p* = 0.712, *p* = 0.235, *p* = 0.218, respectively, data not shown). In addition, the expression of VEGFR-2 in TILs was associated with high serum VEGF-A (Figure 4D) and VEGFR-1 (Figure 4E) levels.

## 3. Discussion

Feline mammary carcinoma shows a highly malignant behavior and a poor prognosis, particularly, the HER2-positive and triple negative subtypes, becoming challenging to treat due to a lack of specific targets [9,30]. Furthermore, angiogenesis is one of the key mechanisms involved in cancer progression, which is controlled by several growth factors secreted by tumor and stromal cells, with VEGF-A being the most potent angiogenic factor [31,32]. Therefore, in this study, the serum levels and tissue expression of VEGF-A and its receptors, VEGFR-1 and VEGFR-2, were evaluated in cats with mammary carcinoma, in order to improve diagnostic tools and therapeutic strategies.

The results showed that serum VEGF-A levels were significantly higher in cats with more aggressive mammary carcinoma subtypes, i.e., HER2-positive and TN normal-like, in accordance with previous studies in human breast cancer [19,33,34,35]. Furthermore, several studies have shown elevated serum VEGFR-1 and VEGFR-2 levels in breast cancer patients, when compared to healthy controls [22,36,37]. Accordingly, the results obtained in this study, revealed that cats showing luminal A, HER2-positive, and TN normal-like tumor subtypes presented higher serum VEGFR-1 and VEGFR-2 levels than control group. This phenomenon might be explained as a compensatory mechanism for high serum VEGF-A levels. Indeed, serum VEGFR-1 and VEGFR-2 receptors can bind to all VEGF-A isoforms, being considered as natural antagonists by decreasing VEGF-A biological activity and its availability for the membrane-bound receptors [13,22,38]. Moreover, a possible reason for the elevated serum VEGFR-1 and VEGFR-2 levels found in cats with luminal A subtype may be related with ulceration. Indeed, all luminal A tumors were ulcerated, suggesting the development of inflammation and consequently the presence of the soluble forms of VEGFR-1 and VEGFR-2, in order to exert anti-inflammatory activities [13]. Furthermore, the results obtained also demonstrated that increased serum VEGFR-1 and VEGFR-2 levels were associated with low-grade tumors, supporting a defense mechanism of these molecules in initial tumor phases against pathological angiogenesis [38]. However, these results were observed in only two animals, with more studies being necessary to better understand this mechanism. Moreover, elevated serum VEGFR-1 and VEGFR-2 levels were also correlated with contraceptive administration. Accordingly, studies in human breast cancer demonstrated that oestrogen and progesterone influence both VEGFR-1 and VEGFR-2 [39,40]. In addition, significant correlations were found between serum VEGF-A levels and serum VEGFR-1 and VEGFR-2 levels, in accordance with that described for human breast cancer [22,36].

The immunohistochemical analysis revealed cytoplasmic immunoreactivity for VEGF-A and VEGFR-1 and cytoplasmic and nuclear staining pattern for VEGFR-2 in cancer cells and TILs, which is consistent with the results of earlier reports [4,14,18,20]. Furthermore, results demonstrated that cats with strong positive VEGF-A expression in TILs showed higher serum VEGF-A levels than cats with a weak positive or negative VEGF-A expression, suggesting an effective endocrine mechanism for the release of serum VEGF-A from stromal cells to the bloodstream. Accordingly, as part of tumor microenvironment, TILs are able to release VEGF-A and inflammatory cytokines, showing immunosuppressive effects [41], including the formation of new blood vessels by acting on endothelial cells [42] and enhancing the inflammatory processes by increasing hypoxia inducible factor 1-alpha (HIF-1α) and VEGF-A synthesis [13]. Moreover, high serum VEGF-A levels were also associated with an intense VEGFR-2 reactivity, suggesting that serum VEGF-A also contributes to the activation of VEGFR-2 [18]. Further, it was identified an association between higher serum VEGFR-1 levels and its weak positive expression in TILs in FMC samples. This finding may be related with VEGFR-1 secretion in tumor microenvironment as a soluble isoform (sVEGFR-1) generated by alternative splicing. Accordingly, Orecchia et al. (2003) demonstrated that sVEGFR-1 present in tumor microenvironment may also play a protumoral action through the stimulation of endothelial cell adhesion and chemotaxis [13,43]. The same could be predicted for VEGFR-2. Finally, sVEGFR-1 can interact with VEGFR-2 abrogating its activity [13]. Whether this provides a compensation mechanism to counteract the concurrently elevated levels of VEGFR-2 that we observed in TILs infiltrating FMCs remains to be established.

## 4. Materials and Methods

### 4.1. Animal Population and Sample Collection

Fifty animals with spontaneous mammary carcinoma that underwent mastectomy and fourteen healthy queens presenting for elective ovariohysterectomy were recruited from Small Animal Hospital of the Faculty of Veterinary Medicine/ULisbon and private clinics around Lisbon. Tumor samples were collected in accordance with the EU Directive 2010/63/EU and all procedures involving the manipulation of animals were consented by the owners. All mammary lesions were embedded in paraffin after fixation in 10% buffered formalin (pH 7.2) during 24–48 h. Serum samples of the same animals were prepared by centrifugation of the fresh blood samples at 1500 g for 20 min at 4 °C and then aliquoted and stored at −80 °C.

For each animal enrolled in the study, the following clinicopathological characteristics were recorded: age, breed, reproductive status, contraceptive administration, treatment performed (none, surgery or surgery plus chemotherapy), number, location and size of tumor lesions, histopathological classification, malignancy grade, presence of tumor necrosis, lymphatic invasion, lymphocytic infiltration, cutaneous ulceration, regional lymph node involvement, stage of the disease (TNM system), DFS and OS. The mean age at diagnosis was 11.8 years (range 7–18 years), while the mean size of the primary lesions was 2.7 cm (range 0.3–7 cm). The DFS was 8.9 ± 1.1 months (*n* = 46; 95% CI: 6.8–11.1 months) and the OS was 13.8 ± 1.3 months (*n* = 49; 95% CI: 11.1–16.5 months).

Regarding the molecular-based subtyping of FMC [8,9], cats were stratified in five groups: Luminal A (*n* = 9), Luminal B (*n* = 17), HER2-positive (*n* = 11), TN Normal-Like (*n* = 5) and TN Basal-Like (*n* = 8).

The homology between human and feline VEGF-A_121,_ VEGF-A_165_ and VEGF-A_165b,_ is 90.8%, 94.2% and 94.1%, respectively (UniProt, accession numbers: *Homo sapiens* P15692-9, P15692-4, P15692-8; *Felis catus* Q95LQ4). Considering the VEGF receptors, the comparison between human and feline VEGFR-1 and VEGFR-2 revealed a homology of 87.8% and 93.2%, respectively (UniProt, accession numbers: *Homo sapiens* P17948, P35968; *Felis catus* M3WIL9, M3WBW2).

### 4.2. Quantification of Serum VEGF-A, VEGFR-1 and VEGFR-2 Levels

The assessment of serum VEGF-A, VEGFR-1 and VEGFR-2 levels was performed using the commercially VEGF (DY293B), VEGF R1/Flt-1 (DY321B) and VEGF R2/KDR (DY357) DuoSet ELISA kits (R&D Systems, Minneapolis, USA), and following the manufacturer’s instructions. The absolute levels of VEGF-A, VEGFR-1 and VEGFR-2 were determined using standard curves (four parameter logistic) run on each ELISA plate. Briefly, the capture antibodies (100 μL/well) were incubated on 96-well plates overnight, at room temperature (RT). On the next day, plates were washed three times (3 × 400 μL/well phosphate buffered saline (PBS)-Tween 0.05%) and coated with 300 μL/well of PBS/BSA blocking agent (1%, *w*/*v*), at RT for 60 min. After another washing step, serum samples previously diluted (1:20) were added (100 μL/well) and incubated during 2 h at RT. Antigen-antibody complexes were washed (3 × 400 μL/well PBS-Tween 0.05%) and 100 μL/well of detection antibodies were added and incubated for 2 h at RT. Then, after three washes with 400 μL/well of PBS-Tween 0.05%, 100 μL/well of streptavidin-HRP were added and incubated for 20 min at RT, avoiding placing the microplate in direct light. Afterwards a further washing step (3 × 400 μL/well PBS-Tween 0.05%), 100 μL/well of substrate solution (1:1 mixture of H_2_O_2_ and tetramethylbenzidine) were added and incubated for 20 min at RT in the dark, followed by a stop solution (50 μL/well of 2 N H_2_SO_4_). A microplate reader was used to measure the optical density at 450 nm and 570 nm (Fluostar Optima Microplate Reader, BMG, Ortenberg, Germany). Standards and negative controls were run on each ELISA plate.

### 4.3. Immunohistochemistry Staining and Evaluation

Immunohistochemistry was done on 3 µm thickness sections of FMC samples (Microtome Leica RM2135, Newcastle, UK). Deparaffinization, rehydration and antigen retrieval were performed using a PT-Link module (Dako, Agilent, Santa Clara, CA, USA), by boiling glass slides in Antigen Target Retrieval Solution pH 9 from Dako, during 20 min at 96 °C. Then, slides were cooled for 30 min at RT and rinsed twice for 5 min in distilled water. Thereafter, sections were blocked with Peroxidase Block Novocastra Solution (Novacastra, Leica Biosystems, Newcastle, UK) during 15 min at RT, followed by two washing steps with PBS pH 7.4, and Protein Block Novocastra Solution (Leica Biosystems) during 10 min. After two washes with PBS for 5 min, tissue slides were incubated with the primary antibodies (Table 2). After incubation, each tissue section was washed with PBS 2x for 5 min and subsquently treated with the Post-Primary Reagent (Leica Biosystems) for 30 min at RT and with the Novolink Polymer (Leica Biosystems) for 30 min. Afterwards, sections were stained with DAB Chromogen Solution (Leica Biosystems) for 5 min and nuclei were counterstained with Gills hematoxylin (Merck, NJ, USA). Slides were dehydrated in an ethanol gradient and mounted with Entellan mounting medium. Human and feline kidney tissues were used as negative and positive controls. Positive and negative control samples were included in each slide run.

The staining of VEGF-A, VEGFR-1 and VEGFR-2 in tumor-infiltrating lymphocytes and cancer cells was assessed manually by two independent pathologists. TILs were identified by their characteristic morphology and scored according to the International TILs Working Group 2014 [44]. Furthermore, cancer cells were evaluated in whole tumor sections with 200–400× magnification. The percentage of positive staining cells was scored using a 4-point scale: 0 (<10%), 1 (10–25%), 2 (26–50%) and 3 (>50%) and the staining intensity was graded as: 0 (no staining), 1+ (weak), 2+ (moderate), and 3+ (strong). The percentage of positive cells and intensity score were then added to obtain a final IHC score [20]. IHC scores of 0–3 were defined as negative, 4–5 as weak positive and 6 as strong positive (Table 3).

### 4.4. Statistical Analysis

Statistical analysis was performed using the SPSS software version 25.0 (IBM, Armonk, NY, USA), while the GraphPad Prism version 8.1.2 (GraphPad Software, CA, USA) was used to plot the graphs. The non-parametric Mann–Whitney U test and the Kruskal–Wallis test were carried out to analyze the differences among groups of continuous variables. Correlations between variables were performed using the Spearman’s rank coefficient. Outliers with more than three standard deviations were removed from analysis. Results with a *p*-value < 0.05 were deemed to be statistically significant.

## 5. Conclusions

In conclusion, cats with HER-2 positive and TN Normal-Like mammary carcinoma subtypes showed more elevated serum VEGF-A, VEGFR-1, and VEGFR-2 levels than healthy animals, suggesting that these molecules may serve as promising non-invasive diagnostic biomarkers for these subtypes. Furthermore, circulating VEGF-A together with its receptors was positively associated with its expression in TILs, indicating that, besides hypoxia, inflammation is another mechanism that leads to cancer progression via VEGF-A/VEGFRs signaling. Altogether, the similarities found between FMC and human breast cancer further validate the utility of the cat as a valuable model for comparative oncology studies.

## Figures and Tables

**Figure 1 cancers-13-00117-f001:**
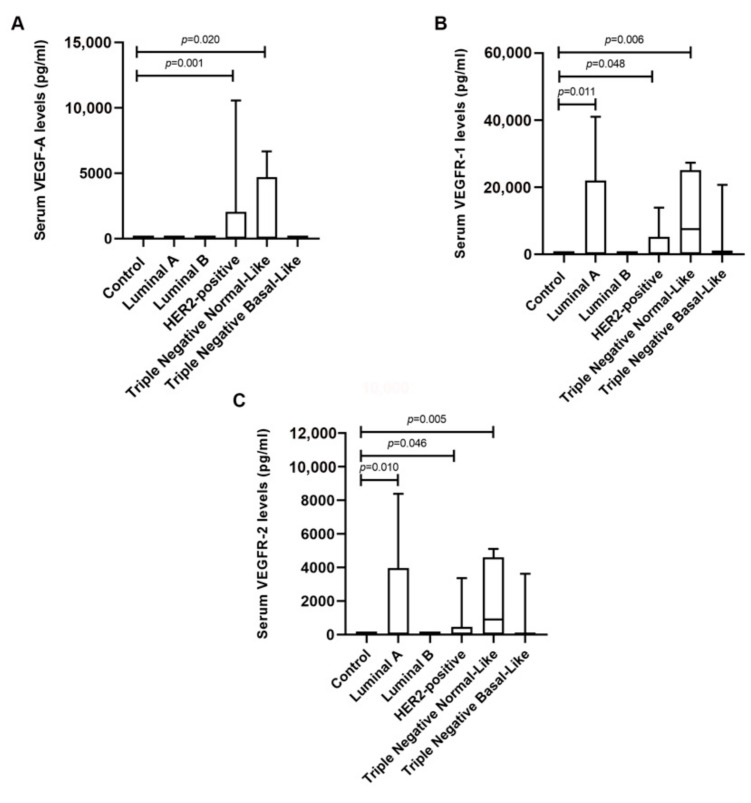
Serum vascular endothelial growth factor A (VEGF-A) levels are significantly increased in cats with HER2-positive and TN Normal-like tumors, while the vascular endothelial growth factor receptor 1 (VEGFR-1) and 2 (VEGFR-2) are significantly elevated in cats with luminal A, HER2-positive and TN Normal-like mammary carcinomas. (**A**) Box plot analysis of serum VEGF-A, (**B**) VEGFR-1 and (**C**) VEGFR-2 levels in the control group and in cats with mammary carcinoma grouped according to their molecular subtype.

**Figure 2 cancers-13-00117-f002:**
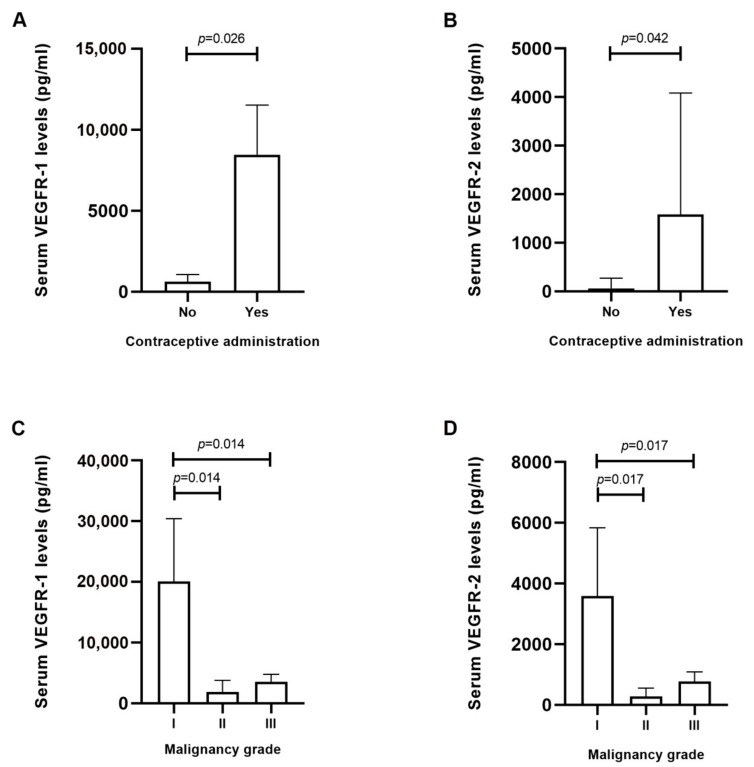
Serum levels of vascular endothelial growth factor receptor 1 (VEGFR-1) and receptor 2 (VEGFR-2) are positively correlated with the use of contraceptives and lower-malignancy tumors. (**A**,**B**) Box-plot analysis showing the mean ± SEM of serum VEGFR-1 and VEGFR-2 levels and its correlation with the use of contraceptive drugs and (**C**,**D**) tumor malignancy grade.

**Figure 3 cancers-13-00117-f003:**
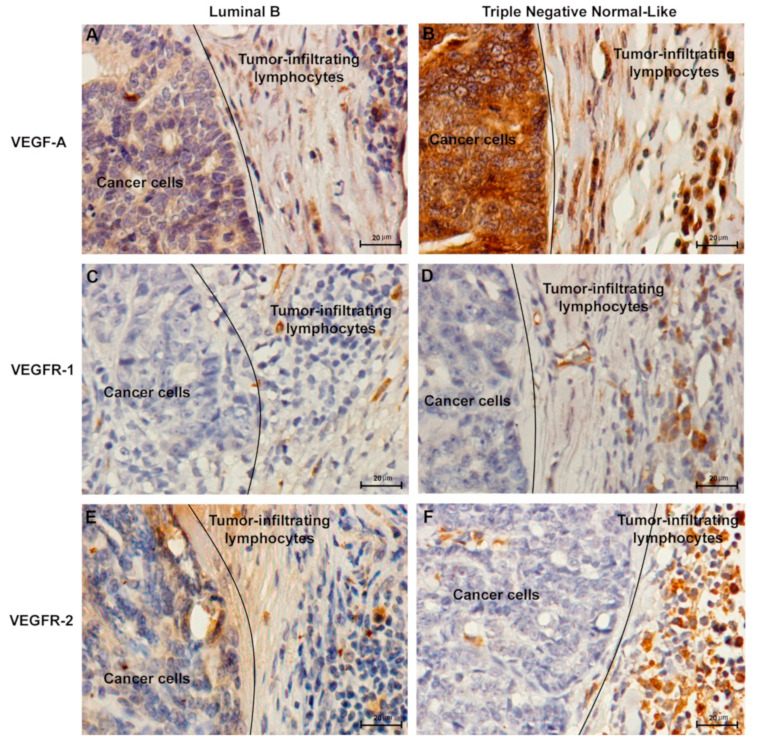
Representative images of immunohistochemical staining of vascular endothelial growth factor (VEGF-A), vascular endothelial growth factor receptor 1 (VEGFR-1) and receptor 2 (VEGFR-2) in tumor infiltrating lymphocytes and cancer cells of feline mammary carcinomas. Luminal B subtype graded as TILs negative for (**A**) VEGF-A, (**C**) VEGFR-1 and (**E**) VEGFR-2. Triple Negative Normal-Like subtype with a TILs-positive score for (**B**) VEGF-A, (**D**) VEGFR-1 and (**F**) VEGFR-2. Original magnification 400×.

**Figure 4 cancers-13-00117-f004:**
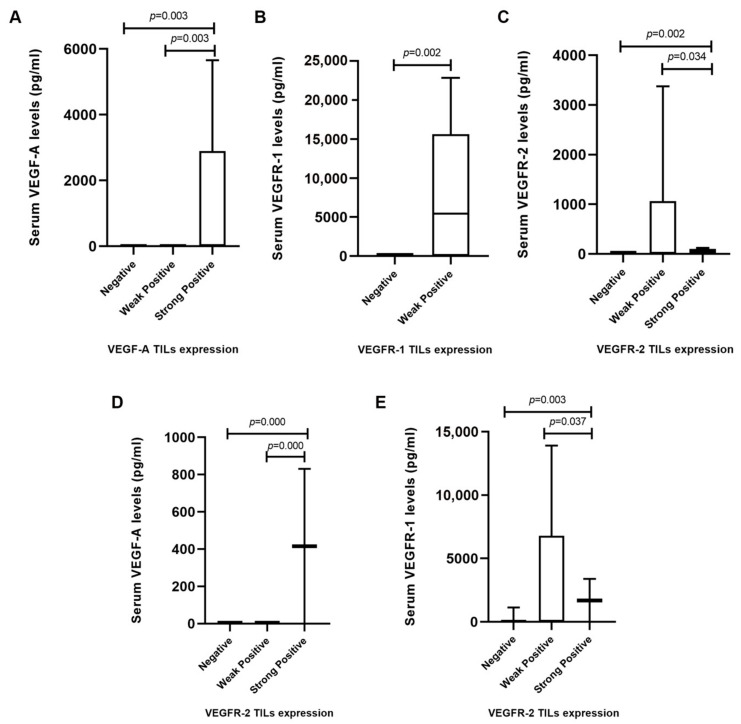
Serum levels and IHC scores of vascular endothelial growth factor (VEGF-A), vascular endothelial growth factor receptor 1 (VEGFR-1) and receptor 2 (VEGFR-2) in tumor infiltrating lymphocytes (TILs) of cats with mammary carcinoma. (**A**) Cats with tumors and a positive score for TILs showed higher serum VEGF-A, (**B**) VEGFR-1 and (**C**) VEGFR-2 levels in comparison with cats that showed a negative score for TILs. (**D**) Box plot diagrams showing that queens with mammary carcinomas scored as VEGFR-2-positive TILs had high serum VEGF-A and (**E**) VEGFR-1 levels.

**Table 1 cancers-13-00117-t001:** Statistical associations between serum VEGF-A, VEGFR-1 and VEGFR-2 levels and clinicopathological parameters examined in cats with mammary carcinoma (mean values ± standard deviation).

Clinicopathological Feature	Number of Animals (%)	VEGF-A (pg/mL)	*p*	VEGFR-1 (pg/mL)	*p*	VEGFR-2 (pg/mL)	*p*
Age			0.483		0.425		0.58
<8 years old	4 (8.0%)	2643.5 ± 5287.0		2442.6 ± 4885.2		337.0 ± 674.1	
8–12 years old	26 (52.0%)	159.5 ± 628.1		3771.3 ± 9414.6		754.1 ± 1838.2	
>12 years old	20 (40.0%)	738.4 ± 2042.5		5565.3 ± 11,514.6		963.3 ± 2288.2	
Spayed			0.075		0.39		0.537
No	24 (48.0%)	1470.3 ± 2914.7		3996.3 ± 8062.8		644.0 ± 1443.7	
Yes	25 (50.0%)	0		5757.9 ± 11,325.2		1117.1 ± 2271.9	
Unknown	1 (2.0%)						
Contraceptive administration			0.188		0.026		0.042
No	21 (42.0%)	660.9 ± 2364.0		1077.3 ± 2740.1		140.4 ± 352.6	
Yes	23 (46.0%)	882.9 ± 1900.5		8156.8 ± 12,291.0		1512.1 ± 2413.4	
Unknown	6 (12.0%)						
Multiple tumors			0.188		0.846		0.701
Negative	19 (38.0%)	476.5 ± 1667.7		6572.4 ± 11,690.2		1217.0 ± 2286.1	
Positive	31 (62.0%)	989.7 ± 2377.6		3602.3 ± 8396.6		621.3 ± 1617.4	
Lymph node status			0.155		0.345		0.432
Negative	31 (62.0%)	1102.7 ± 2557.0		4817.9 ± 9291.5		840.3 ± 1742.9	
Positive	16 (32.0%)	0		4842.9 ± 10,931.0		929.7 ± 2235.5	
Unknown	3 (6.0%)						
Stage			0.502		0.606		0.688
I	11 (22.0%)	1753.6 ± 3736.3		4766.1 ± 8059.3		882.4 ± 1572.0	
II	7 (14.0%)	138.5 ± 339.3		5632.5 ± 11,354.2		993.5 ± 2198.0	
III	27 (54.0%)	387.6 ± 1312.0		3609.1 ± 10,349.2		663.2 ± 2051.1	
IV	5 (10.0%)	0		6914.8 ± 10,335.0		1213.0 ± 1774.6	
Tumor size			0.67		0.374		0.5
≤2 cm	26 (52.0%)	835.0 ± 2604.1		6024.2 ± 11,286.1		1140.8 ± 2239.7	
>2 cm	24 (48.0%)	467.9 ± 1405.1		2754.6 ± 7646.5		452.9 ± 1386.5	
Tumor malignancy grade			0.198		0.037		0.046
I	2 (4.0%)	5286.9 ± 7476.9		20,094.3 ± 14,600.2		3591.6 ± 3172.7	
II	6 (12.0%)	0		1899.8 ± 4653.5		278.8 ± 683.0	
III	42 (84.0%)	480.0 ± 1526.4		3278.2 ± 9626.1		776.5 ± 1888.2	
Tumor necrosis			0.587		0.227		0.182
Negative	11 (22.0%)	1358.2 ± 3725.1		8079.7 ± 14,010.3		1640.2 ± 2907.7	
Positive	39 (78.0%)	415.3 ± 1549.8		2801.0 ± 8408.6		461.7 ± 1551.2	
Tumor lymphatic invasion							
Negative	43 (86.0%)	544.3 ± 2112.3	0.956	4537.3 ± 10,320.0	0.098	820.5 ± 2011.0	0.117
Positive	7 (14.0%)	941.9 ± 2307.1		0		0	
Lymphocytic infiltration			0.466		0.316		0.523
Negative	16 (32.0%)	881.2 ± 2932.7		5173.4 ± 9837.5		901.9 ± 1818.4	
Positive	33 (66.0%)	485.1 ± 1669.0		3292.4 ± 9802.5		609.8 ± 1949.4	
Unknown	1 (2.0%)						
Tumor ulceration			0.073		0.116		0.094
Negative	43 (86.0%)	682.0 ± 2286.4		3720.8 ± 9483.9		704.6 ± 1861.9	
Positive	7 (14.0%)	161.5 ± 1020.3		4626.6 ± 11,316.9		656.7 ± 2151.0	
Metastasis							
No	22 (44%)	535.5 ± 198.8	0.89	5740.6 ± 11,041.4	0.165	1093.1 ± 2233.9	0.269
Yes	28 (56%)	747.6 ± 2810.5		3412.8 ± 8447.5		595.1 ± 1526.0	

**Table 2 cancers-13-00117-t002:** Primary antibodies and their conditions of use.

Monoclonal Antibody	Reference	Dilution	Incubation Time and Temperature
Anti-VEGF	Clone VG1 (Novus Biologicals)	1:50	60’ at RT
Anti-VEGFR1/Flt-1	Clone CL0345 (Novus Biologicals)	1:200	60’at RT
Anti-VEGFR2/KDR/Flk-1	Clone EIC (Novus Biologicals)	1:10	120’ at RT plus 4 °C overnight

RT—Room Temperature.

**Table 3 cancers-13-00117-t003:** Scoring criteria of immunostaining assay for VEGF-A, VEGFR-1 and VEGFR-2.

Percentage of Positive Staining Cells		Staining Intensity	
Score	Interpretation	Score	Interpretation
0	<10%	0	No staining
1	10–25%	1	Weak
2	26–50%	2	Moderate
3	>50%	3	Strong
**Total score (0–6): Score of positive staining cells + intensity score**			
0–3: Negative			
4–5: Weak Positive			
6: Strong Positive			

## Data Availability

The data presented in this study are available on request from the corresponding author. The data are not publicly available as their containing information that could compromise the privacy of future research.

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
