# Peer review of "Diagnostic Value of VEGF-A, VEGFR-1 and VEGFR-2 in Feline Mammary Carcinoma"

_cancers, 2021, doi:10.3390/cancers13010117_

Round 1

Reviewer 1 Report

The authors aimed to assess the potential of VEGF-A/VEGFRs pathway in FMC. Their results showed that serum VEGF-A, VEGFR-1, and VEGFR-2 levels were significantly higher in cats with HER2-positive and TN normal-like tumors, presenting a positive association with its tumor-infiltrating lymphocytes expression.

The results are very interesting, but I still think there are some findings that must be addressed prior to publication

Results- Line 104 to 119

2.1. Serum VEGF-A, VEGFR-1, and VEGFR-2 levels are significantly elevated in cats with HER2-positive 106 and TN Normal-Like mammary carcinoma.

It would be important to have the SD associated with the average.

Fig 1 B- What is the average of the Triple Negative Basal-Like? How is it possible to have such a low average near 0 with such a high SD?

Line 134

Table 1- The SD deviations should also be added and not just the average

Table 1-It would be nice to have in the first column, how many animals exist per clinicopathological item, instead of having those values in the material and methods section. It would be easy to read if they were here at just one table.

Table 1- Could the authors clarify the differences between Lymph node status and metastasis? And why so different results between the 2.

Fig 3-Overall the pictures were taken at a very low magnification to allow the reader to do their own interpretation. For instance in the Triple Negative Basal Like VEGF-A picture, the immunostaining shows too much background. Both the stromal cells and the TILS all seem to have staining. I would like to have other pictures at higher magnification.

The same for the Triple Negative Basel Like VEGFR-1 picture.  The magnification is very low, and it is impossible to see which cells are stained or not.  

Line 182 The authors showed that serum VEGF-A levels were significantly higher in cats with more aggressive mammary carcinoma subtypes, i.e. HER2-positive and TN normal-like.

How do authors explain that VEGF-A levels are higher in more aggressive tumors, but on the other hand are higher in grade I tumors compared to grade III and not present at all in grade II (even if with no statistically significant differences)? Shouldn’t we see higher serum VEGF-A levels in tumors with lymphatic invasion, metastasis, Grade III malignancy? Or the increase VEGF-A in HER2-positive and TN normal-like carcinomas is associated with the subtype by other reasons than malignancy.

The authors also state that the results obtained demonstrated that increased serum VEGFR-1 and VEGFR-2 levels were associated with low-grade tumors, supporting a defense mechanism of these molecules in initial tumor phases against pathological angiogenesis [40]. There are only 2 animals with grade I tumors. I would like to have more tumors in this grade to advance with this observation.

The authors also say that a possible reason for the elevated serum VEGFR-1 and VEGFR-2 levels found in cats with luminal A subtype may be related to contraceptive administration. However, it was not clear to me that, in this study, the animals that received contraceptives had more luminal A subtype tumors. And, If this was the reason, shouldn’t all cats that received contraceptives had high serum VEGFR-1 and VEGFR-2 levels independently of the tumor subtype. 

The authors also say that a positive correlation was identified between serum VEGF-A and VEGFR-1 and VEGFR-2 levels but no VEGF-A was found in luminal A in spite of the highest levels in VEGFR-1 and VEGFR-2. How do the authors explain that?

In figure 3, It seems there is some IHC staining for VEGF-A in the tumor cells. But apparently, no VEGF-A was found in the serum. Can the authors explain this finding?

In material and methods, I would like that the authors explained how the IHC staining was done for the cancer epithelial cells. In how many fields and at what magnification it was done?

Author Response

The authors aimed to assess the potential of VEGF-A/VEGFRs pathway in FMC. Their results showed that serum VEGF-A, VEGFR-1, and VEGFR-2 levels were significantly higher in cats with HER2-positive and TN normal-like tumors, presenting a positive association with its tumor-infiltrating lymphocytes expression.

The results are very interesting, but I still think there are some findings that must be addressed prior to publication. Dear reviewer, thank you very much for your insightful comments and constructive remarks along the manuscript. They really improve the final quality of the manuscript. 

Results- Line 104 to 119

2.1. Serum VEGF-A, VEGFR-1, and VEGFR-2 levels are significantly elevated in cats with HER2-positive and TN Normal-Like mammary carcinoma.

It would be important to have the SD associated with the average. As requested, SD values were presented between lines 111-117.

Fig 1 B- What is the average of the Triple Negative Basal-Like? How is it possible to have such a low average near 0 with such a high SD? Dear reviewer, the statistical analysis showed a mean value ± SD of 3127± 7780.1 pg/ml for cats with Triple Negative Basal-Like mammary carcinoma. Of the nine cats with this tumor subtype, seven presented mean values of 0.0 pg/ml.

Line 134

Table 1- The SD deviations should also be added and not just the average. As suggested, SD deviations were added in Table 1 and mentioned in caption of Table 1 (lines 136-137).

Table 1-It would be nice to have in the first column, how many animals exist per clinicopathological item, instead of having those values in the material and methods section. It would be easy to read if they were here at just one table. As requested, an additional column was added in Table 1 with the number of animals that exist per clinicopathological feature. Therefore, the Table 2 was removed and the number of tables have been updated accordingly (lines 289, 296, 307 and 308).

Table 1- Could the authors clarify the differences between Lymph node status and metastasis? And why so different results between the 2. While the lymph node status identifies animals that only showed metastasis in regional lymph nodes, metastasis include cats with both regional and distant metastasis, meaning more animals in this feature and therefore distinct values.

Fig 3-Overall the pictures were taken at a very low magnification to allow the reader to do their own interpretation. For instance in the Triple Negative Basal Like VEGF-A picture, the immunostaining shows too much background. Both the stromal cells and the TILS all seem to have staining. I would like to have other pictures at higher magnification. The same for the Triple Negative Basel Like VEGFR-1 picture.  The magnification is very low, and it is impossible to see which cells are stained or not.  Dear reviewer, a new figure 3 was prepared with a higher magnification (400x), in order to show the staining pattern on TIL’s and cancer cells. Thank you for this insightful suggestion.

Line 182 The authors showed that serum VEGF-A levels were significantly higher in cats with more aggressive mammary carcinoma subtypes, i.e. HER2-positive and TN normal-like.  How do authors explain that VEGF-A levels are higher in more aggressive tumors, but on the other hand are higher in grade I tumors compared to grade III and not present at all in grade II (even if with no statistically significant differences)? Shouldn’t we see higher serum VEGF-A levels in tumors with lymphatic invasion, metastasis, Grade III malignancy? Or the increase VEGF-A in HER2-positive and TN normal-like carcinomas is associated with the subtype by other reasons than malignancy. Dear reviewer, thank you for this relevant remark. As you mentioned, no statistically significant associations were found between serum VEGF-A levels and the recorded clinicopathological features. Indeed, the higher serum VEGF-A levels found in the two animals with grade I tumors may be considered a statistical artifact, as one of the animals presented a mean value of 0.0 pg/ml.

The authors also state that the results obtained demonstrated that increased serum VEGFR-1 and VEGFR-2 levels were associated with low-grade tumors, supporting a defense mechanism of these molecules in initial tumor phases against pathological angiogenesis [40]. There are only 2 animals with grade I tumors. I would like to have more tumors in this grade to advance with this observation. Dear reviewer, although we agree with your comment, we would like to remind you that this work results from a follow-up study of 5 years that enrolled queens with mammary carcinoma. However, as you suggested, authors discussed a bit more this issue in lines200-201. Thank you for this suggestion.

The authors also say that a possible reason for the elevated serum VEGFR-1 and VEGFR-2 levels found in cats with luminal A subtype may be related to contraceptive administration. However, it was not clear to me that, in this study, the animals that received contraceptives had more luminal A subtype tumors. And, If this was the reason, shouldn’t all cats that received contraceptives had high serum VEGFR-1 and VEGFR-2 levels independently of the tumor subtype. Dear reviewer, thank you for raising this very interesting point. In fact, further data analysis allowed us to realize that all luminal A tumors presented ulceration. Therefore, authors rephrased lines 193-197 and lines 201-202 in the discussion section.

The authors also say that a positive correlation was identified between serum VEGF-A and VEGFR-1 and VEGFR-2 levels but no VEGF-A was found in luminal A in spite of the highest levels in VEGFR-1 and VEGFR-2. How do the authors explain that? Dear reviewer, although positive correlations were found between serum VEGF-A and both VEGFR-1 and VEGFR-2 levels, the Spearman’s correlation coefficients only showed moderate values (r=0.567; r=0.591), as stated in lines 119-120.

In figure 3, It seems there is some IHC staining for VEGF-A in the tumor cells. But apparently, no VEGF-A was found in the serum. Can the authors explain this finding? Dear reviewer, authors didn’t find any significant associations between serum VEGF-A levels and its expression in cancer cells, as mentioned in lines 166-167.

In material and methods, I would like that the authors explained how the IHC staining was done for the cancer epithelial cells. In how many fields and at what magnification it was done? As requested, additional information was added in line 302. The authors would like to thank you once again for your suggestions that greatly improved this work.

Reviewer 2 Report

The manuscript submitted for publication “Elevated serum
VEGF-A, VEGFR-1 and VEGFR-2 levels are associated with aggressive feline mammary carcinoma subtypes, being correlated
with their expression in tumor infiltrating lymphocytes
” reports a study of high importance regarding feline mammary
tumors.
However, despite the relevance that I recognize in the work there are some questions to the authors: The authors state in the discussion (lines 193 to 195) that a possible reason for the high serum levels of VEGFR-1 and
VEGFR-2 recorded in cats presenting Luminal A subtype carcinoma may be related to the use of contraceptives. In Table 2, the number of animals presented with contraceptive administration is 23 (46.0%), however only 9 animals are classified as Luminal A group. Is there any other associated factor? If so, please consider it in the discussion session.
In the material and methods section the authors describe theELISA technique (line 251) for the cat’s serum measurement
of VEGF-A, VEGFR-1 and VEGFR-2, using a commercial kit (DuoSet ELISA kit) that is produced for humans. Even taking into account that the homology described for VEGFR-1 and VEGFR-2 of the cat is 87.8% and 93.25 respectively
(lines 246 and 247) it is assumed that a positive control
was included in the test. It should be included in the text.
Last but not least, I suggested that the title of the
article be shortened as much as possible.

Author Response

The manuscript submitted for publication “Elevated serum VEGF-A, VEGFR-1 and VEGFR-2 levels are associated with aggressive feline mammary carcinoma subtypes, being correlated with their expression in tumor infiltrating lymphocytes” reports a study of high importance regarding feline mammary tumors. Dear reviewer, thank you for your comments and suggestions. All the authors appreciate your valuable review.

However, despite the relevance that I recognize in the work there are some questions to the authors: 

The authors state in the discussion (lines 193 to 195) that a possible reason for the high serum levels of VEGFR-1 and VEGFR-2 recorded in cats presenting Luminal A subtype carcinoma may be related to the use of contraceptives. In Table 2, the number of animals presented with contraceptive administration is 23 (46.0%), however only 9 animals are classified as Luminal A group. Is there any other associated factor? If so, please consider it in the discussion session. Dear reviewer, thank you a lot for this relevant remark. In fact, further data analysis allowed us to realize that all luminal A tumors presented ulceration. Therefore, authors rephrased lines 193-197 and lines 201-202 in the discussion section.

In the material and methods section the authors describe the ELISA technique (line 251) for the cat’s serum measurement of VEGF-A, VEGFR-1 and VEGFR-2, using a commercial kit (DuoSet ELISA kit) that is produced for humans. Even taking into account that the homology described for VEGFR-1 and VEGFR-2 of the cat is 87.8% and 93.25 respectively (lines 246 and 247) it is assumed that a positive control was included in the test. It should be included in the text. Dear reviewer, although we agree with your comment, to the best of our knowledge this is the first study that evaluates the serum VEGF-A, VEGFR-1 and VEGFR-2 levels in cats stratified by mammary carcinoma molecular subtype. Unfortunately, we didn’t have any previous confirmed positive serum sample of a cat with mammary carcinoma. Nevertheless, standards and negative controls were used on each ELISA plate (lines 278 and 279). Thank you for this insightful suggestion.

Last but not least, I suggested that the title of the article be shortened as much as possible. As suggested, the title was modified to “Diagnostic value of VEGF-A, VEGFR-1 and VEGFR-2 in feline mammary carcinoma”.

Reviewer 3 Report

The authors address their work to the usefulness of VEGF-A and its serum receptors as drug targets in cats affected by FCM. The paper is well documented, rich of information and has the overall merit of digging in the VEGF/VEGFRs pathway in feline mammary cancer as potential comparative model.

Still, I have some concerns: 

Please correct references all over the text since often are not pertinent to the sentence. For example, papers n°15-17 are not about TILs and VEGF but refer only to serum levels of VEGF in breast cancer patients. Line 51: at the end of the sentence the reference is lacking; the above reported papers by Millanta and Zappulli are not about the molecular profile of FCM.

Paragraph 2.3. In fig. 3 A, C, E immunopositivity for TILs can not be evaluated. In fig. 3 C, E cancer cells are negative.  In fig. 3 F looks like only immune cells are positive both intratumoral and peritumoral.

Line 218: sVEGFR in TILs?

Author Response

The authors address their work to the usefulness of VEGF-A and its serum receptors as drug targets in cats affected by FCM. The paper is well documented, rich of information and has the overall merit of digging in the VEGF/VEGFRs pathway in feline mammary cancer as potential comparative model. Dear reviewer, thank you so much for your positive opinion about this manuscript, we appreciate your very valuable review.

Still, I have some concerns: 

Please correct references all over the text since often are not pertinent to the sentence. For example, papers n°15-17 are not about TILs and VEGF but refer only to serum levels of VEGF in breast cancer patients. Line 51: at the end of the sentence the reference is lacking; the above reported papers by Millanta and Zappulli are not about the molecular profile of FCM. Dear reviewer, thank you for this relevant remark. The references were confirmed and, as recommended, in line 53, two references were added at the end of the sentence. Furthermore, the references of reported papers by Millanta and Zappulli were removed from the article.

Paragraph 2.3. In fig. 3 A, C, E immunopositivity for TILs can not be evaluated. In fig. 3 C, E cancer cells are negative.  In fig. 3 F looks like only immune cells are positive both intratumoral and peritumoral. Dear reviewer, considering the figure 3 and also the suggestion of reviewer #1, new imagens were added with a higher magnification (400x), in order to show the differences among tumors graded as TILs negative (Luminal B tumors – Fig. 3 A, C, E) and tumors with immunopositivity for TILs (Triple Negative Normal-like tumors - Fig 3. B, D, F). Thank you, a lot, for this suggestion.

Line 218: sVEGFR in TILs? Corrected (line 228).